# Seminal-Plasma-Mediated Effects on Sperm Performance in Humans

**DOI:** 10.3390/cells11142147

**Published:** 2022-07-08

**Authors:** Tanja Turunen, Martina Magris, Marjo Malinen, Jukka Kekäläinen

**Affiliations:** Department of Environmental and Biological Sciences, University of Eastern Finland, P.O. Box 111, 80101 Joensuu, Finland; martina.magris@gmail.com (M.M.); marjo.malinen@uef.fi (M.M.); jukka.s.kekalainen@uef.fi (J.K.)

**Keywords:** fertilization, infertility, protein, reproduction, seminal plasma, sperm

## Abstract

Seminal plasma (SP) plays a crucial role in reproduction and contains a large number of proteins, many of which may potentially modify sperm functionality. To evaluate the effects of SP identity and its protein composition on human sperm function, we treated the sperm of several males with either their own or multiple foreign SPs in all possible sperm–SP combinations (full-factorial design). Then we recorded sperm motility and viability in these combinations and investigated whether the sperm performance is dependent on sperm and SP identity (or their interaction). Finally, we studied whether the above-mentioned sperm traits are affected by the abundance of three SP proteins, dipeptidyl peptidase-4 (DPP4), neutral endopeptidase (NEP), and aminopeptidase N (APN). The identity of the SP donor affected sperm swimming velocity, viability, and the proportion of hyperactivated sperm, but males’ own SP was not consistently more beneficial for sperm than foreign SPs. Furthermore, we show that sperm performance is also partly affected by the interaction between sperm and SP donor. Finally, we found that DPP4 and NEP levels in SP were positively associated with sperm swimming velocity and hyperactivation. Taken together, our results highlight the importance of seminal plasma as a potential source of biomarkers for diagnostics and therapeutic interventions for male-derived infertility.

## 1. Introduction

Seminal plasma (SP), the acellular fraction of the semen, is a complex mixture of components secreted from the epididymis and the accessory sexual glands. Originally, SP was thought to function only as a vehicle for spermatozoa to enter the female reproductive system [1]. More recent studies have, however, demonstrated that SP has a multifunctional role in the fertilizing process (reviewed by [2,3]). Accordingly, SP is nowadays recognized as a crucial factor in many aspects of male reproduction, modulating sperm functionality and phenotype, and even affecting offspring performance [4,5,6]. Nevertheless, the overall understanding of how SP factors control human sperm kinetics is limited.

Some animal studies have shown that sperm supplementation with SP before insemination is beneficial since it increases the ability of sperm to penetrate the cervical mucus, thereby improving the probability of pregnancy [7]. It has also been demonstrated that SP influences sperm performance, especially sperm swimming speed and viability [8,9]. Furthermore, two studies, performed on stallions and rams, have suggested that the effect of SP supplementation on sperm traits depends on the origin of both the SP and the sperm [10,11]. Despite these findings, the physiological significance of SP for fertilization success, especially in humans, has often been questioned as the exposure of sperm to SP seems to be unnecessary for the sperm fertilization capacity [12]. Similarly, the functional importance of SP has largely been omitted in current male infertility diagnostics, which are almost exclusively based on various sperm traits, such as sperm concentration, motility, and morphology, i.e., the cellular fraction of the semen [13].

This approach may not be entirely warranted as SP also serves as a rich source of biomarkers for male-derived infertility [14,15], which could potentially help to improve the accuracy of male infertility diagnostics. However, modern in vitro fertilization (IVF) and intracytoplasmic sperm injection (ICSI) techniques can often overcome the challenges caused by sperm abnormalities, which has hindered the development of male infertility treatments [16]. In other words, even in male-factor infertility cases, the treatment burden is currently fallen mainly on the female partner [16]. To equalize this burden, more research is needed to better understand the role of various male-derived factors in the modulation of sperm performance. Therefore, a deeper understanding of SP factors could pave the way for the development of treatment approaches that would directly improve sperm performance and in this way help to reduce women’s health risks associated with invasive infertility treatments.

An increasing number of studies have focused on investigating the functions and abundance of individual SP proteins [17,18,19]. For example, Jodar et al. identified 380 SP proteins that are present in sperm only after SP exposure, and that thus seems to be actively incorporated from the SP into the sperm [20]. The functional analysis of these SP proteins revealed, inter alia, peptidase activity that is required, for example, for sperm motility [20]. Previous studies have experimentally illustrated that two SP peptidases, dipeptidyl peptidase-4 (DPP4) and aminopeptidase N (APN), are transferred in sperm via SP extracellular vesicles [21,22]. Furthermore, APN and another peptidase, neutral endopeptidase (NEP), have been hypothesized to be involved in the control of sperm performance [23]. Peptidase activity generally refers to the enzymes that are involved in the degradation of particular signaling molecules, such as endogenous opioids in the case of NEP. The presence of endogenous opioids and the activation of opioid receptors has been previously linked to the inhibition of sperm’s fertilization capacity, especially via decreased sperm motility [24,25,26,27]. However, the detailed mechanisms of action of SP proteins in the control of sperm functionality remain to be determined.

In the present study, we hypothesized that SP peptidases are involved in the control of sperm performance. We expect that identifying such functionally important proteins can facilitate further studies aiming to develop new biomarkers for male-factor infertility and open novel possibilities to treat male infertility. We performed a full-factorial experiment, where we treated the sperm of nine males with both their own and eight foreign SPs in all possible (*n* = 81) combinations and studied how male-specific variation in SP and its protein composition affected sperm performance. We measured sperm swimming velocity, hyperactivation, the proportion of motile sperm, and viability in all these combinations at two time points (30 min and 90 min after the initiation of SP treatments). Furthermore, we quantified the relative expression of three proteins (DPP4, NEP, and APN) in all SP samples using Western blot analysis. Our subsequent analyses revealed positive associations between DPP4 and NEP levels in SP and sperm curvilinear swimming velocity and hyperactivation.

## 2. Materials and Methods

### 2.1. Sample Collection and Separation of Sperm and SP

Participants (*n* = 9 males, mean age 28.5 ± 1.2 S.E. years) were recruited via public advertisements or from the fertility clinic of the North Karelia Central Hospital, Finland. All the subjects signed informed consent prior to sample donation. All the subjects were healthy, Caucasian males, and none of them smoked tobacco. All the semen samples were of normal quality in terms of sperm count (>15 million cells/mL), liquefaction time (<60 min), and viscosity (<2 cm) according to WHO’s criteria [13]. Exclusion criteria included semen volume less than 1.5 mL and a history of male-factor infertility. Semen samples were provided by masturbation after two to seven days of sexual abstinence. Each male provided two samples, with an interval of 16–41 days between each other. The first sample was used to obtain and store the SP, and to measure sperm performance in each man’s own seminal fluid. The second sample was used to isolate sperm cells and measure their performance in all possible sperm–SP combinations.

Freshly ejaculated samples were kept at 37 °C for 30 min to allow semen to liquefy. Then sperm samples were centrifuged at 16,000× *g* for 20 min; the supernatant (SP) was collected and centrifuged again for a further 20 min. After the second centrifugation, the supernatant was inspected under the microscope (negative phase contrast objective, 100× *g* magnification) to ensure that no sperm cells were present. Finally, supernatants were aliquoted and stored at −80 °C in individual (male-specific) tubes.

After liquefaction (see above) the second semen samples were split into two to four 500 μL aliquots (depending on the total volume of semen) that were further diluted with 1000 μL of PureSperm^®^ Wash solution (Nidacon International AB, Mölndal, Sweden). These diluted samples were centrifuged at 500× *g* for 10 min. The supernatant was then removed, and the sperm pellet was resuspended in 400 μL of PureSperm^®^ Wash solution and centrifuged at 500× *g* for a further 10 min. Sperm cells obtained from individual aliquots were pooled and resuspended in PureSperm^®^ Wash solution to reach a final concentration of 70 million sperm mL^−1^.

### 2.2. Treatment of Sperm with SP

On the day of each man’s first semen donation, previously separated sperm and own seminal plasma (see above) were re-combined 1:1 (25 µL sperm solution + 25 µL SP) in two separate subsamples (A and B). On the day of each male’s second semen donation, one tube of frozen SP from all the males was thawed and split into two sub-samples (A and B). Washed sperm aliquots from each male were then individually combined (1:1, 25 µL sperm solution + 25 µL SP) with both SP sub-samples, in all possible sperm–SP combinations and with PureSperm^®^ Wash solution (hereafter called as SW) that was considered as a control sample. This full-factorial design yielded 180 sperm–SP combinations (9 sperm donors × 9 SP donors = 81 sperm–SP combinations and 9 control samples, 2 sub-samples of each).

### 2.3. Sperm Motility and Viability Measurements

Sperm performance was measured in PureSperm^®^ Wash solution (SW), in own SP, and in eight foreign SP samples. To measure sperm motility parameters, we transferred 1 µL of each SW- and SP-treated sperm sample to pre-warmed Leja 4-chamber microscope slides (Leja, Nieuw-Vennep, The Netherlands). Then, we used computer-assisted sperm analysis (CASA; Integrated Semen Analysis System, ISAS v. 1.2 Proiser, Valencia, Spain), with a negative phase-contrast objective (100× *g* magnification) and a capture rate of 100 frames s^−1^ to measure sperm motility (curvilinear velocity: VCL; linearity: LIN; the amplitude of lateral head displacement: ALH; and the percentage of motile sperm). Sperm motility was measured from two replicate recordings for each sub-sample, resulting in 360 (90 combinations × 2 sub-samples × 2 replicates) recordings in total. All the sperm traits were recorded at two time points: 30 min and 90 min since the beginning of the SP treatment. Sperm motility was measured on average from 120 ± 5 sperm cells (mean ± SE, range 62–201) per replicate. The percentage of hyperactivated sperm was determined based on the following three CASA parameters: VCL > 150 μm s^−1^, LIN < 50%, and ALH > 2.0 [28]. Sperm VCL and the proportion of motile cells in the presence of own SP were strongly correlated between the two sampling periods (Pearson correlation analysis, VCL 30 min: r = 0.71, *p* = 0.033; VCL 90 min: r = 0.91, *p* = 0.001; motility 30 min: r = 0.50, *p* = 0.17; motility 90 min: r = 0.67, *p* = 0.048, *n* = 9, in all cases).

At the end of the motility measurements, 5 µL from each SW- and SP-treated subsample was diluted in PureSperm^®^ Wash solution to a final volume of 25 µL and then stained with 0.6 µL of propidium iodide (PI, 5 µg/mL, Thermo Fisher, Eugene, OR, USA). After three minutes of incubation in the dark, 0.5 µL of 1% formalin was added to immobilize the sperm and 10 µL of each PI-treated subsample was individually transferred to a LUNA™ Reusable Slide (Logos Biosystems, Annandale, VA, USA). The number of dead and total sperm cells were counted using a LUNA-FL™ Dual Fluorescence Cell Counter (Logos Biosystems, Annandale, VA, USA). Sperm viability was calculated as the percentage of live spermatozoa (not PI-stained) of the total number of counted spermatozoa. All viability measurements included two replicate recordings for each SP sub-sample, resulting in 360 recordings (81 combinations + 9 controls × 2 sub-samples × 2 replicates) in total. Sperm viability was measured on average from 4532 ± 57 sperm cells (mean ± SE, range 1193–8038) per replicate.

To account for the time needed for sperm motility and sperm viability measurements, the initiation of the SP treatments was performed at three minutes intervals between the subsamples. This way, each subsample was measured after an identical interval from the beginning of the treatment. In addition, to minimize the potential time effect on measured sperm traits, sperm motility and viability in the first sub-sample (A) were always measured in the following order: SP1, SP2,…, SP9, whereas in the second sub-sample (B) measurements were performed in the opposite order: SP9, SP8,…, SP1. All sperm measurements were always performed by using fresh sperm on the day of the second semen collection.

### 2.4. Western Blot Analyses of Protein Levels

To balance the protein amount between gel wells for Western blot, the protein content of each seminal plasma sample was measured using Pierce™ BCA Protein Assay Kit (Thermo Scientific, Rockford, IL, USA). Then, equal amounts of protein (15 μg) from each sample were denatured with 5× SDS sample buffer at 100 °C for 5 min and separated using Mini-PROTEAN^®^ TGX Stain-Free™ Precast Protein Gels (Bio-Rad, Hercules, CA, USA). Precision Plus Protein™ WesternC™ Blotting Standard (Bio-Rad) was used as a molecular weight standard. After electrophoresis, proteins in the gels were imaged according to the manufacturer’s protocol using the ChemiDoc System (Bio-Rad). Proteins were transferred to a polyvinyl difluoride (PVDF) membrane (Bio-Rad) using a Trans-Blot^®^ Turbo™ Transfer System (Bio-Rad) and membranes were subsequently blocked for 1 h in saturation buffer (2% skimmed milk, 10 mM Tris, 0.14 M NaCl, pH 7.4) at RT. Blots were incubated at 4 °C overnight with primary antibodies for ANP (1:2000, ab108310, Abcam, Amsterdam, Netherlands), DPP4 (1:1000, AF1180, R&D systems, Minneapolis, MN, USA), and NEP (1:1000, AF1182, R&D systems). Then, blots were incubated for 45 min with secondary antibodies at RT: HRP-conjugated anti-rabbit IgG (1:2000, G-21234, Invitrogen) for APN and anti-goat IgG (1:1000, HAF017, R&D systems) for DPP4 and NEP. Prior to detection with the ChemiDoc System, blots were incubated in Pierce ECL Western Blotting Substrate (Thermo Scientific) for 5 min at RT. Analyses of gel and blot images were performed using the Image Lab Software (Bio-Rad). Utilized stain-free gel technology allows compensation of loading variations using standardized total protein normalization, i.e., the intensity of protein of interest is normalized to the corresponding total protein intensity [29]. The average expression of each protein was densitometrically calculated based on three technical replicates and the average value was used for subsequent analysis.

### 2.5. Functional Enrichment Analysis

We performed functional enrichment analysis for NEP, DPP4, and APN (Homo sapiens) and their interactive proteome based on the STRING database [30]. We inputted all three proteins into “multiple protein” search and visualized the full STRING network with functional and physical associations based on the following search criteria: “textmining”, “co-expression”, “experiments”, and “databases”. We allowed a maximum of 10 interactions of the 1st shell and 20 interactions in the 2nd shell (*n* = 33 proteins in total). All interactions had to fulfill the highest confidence score (0.900). The top 20 gene ontology terms of biological processes according to strength value were plotted and ordered by −log(false discovery rate).

### 2.6. Statistical Analyses

Linear mixed models (LMM) and generalized linear mixed models (GLMM) were used in R (v. 3.6.2) to test the effects of different SP treatments on sperm parameters [31]. The effects of treatments on sperm swimming velocity (VCL) were tested in LMM with the lmer function (package lme4 v. 1.1-28) [32]. Model assumptions were graphically verified using Q-Q plots and residual plots. The *p*-values for both fixed and random effects in all LMMs were computed with anova and ranova functions (package lmerTest v.3.1-3), respectively [33]. The effects of treatments on the proportions of motile and hyperactivated sperm as well as sperm viability were tested in GLMM with a binomial distribution (each of these in a separate model) with the glmer function (package lme4 v. 1.1-28) [32]. The number of motile sperm, hyperactivated sperm, or viable sperm, and the total number of sperm were implemented as two-vector response variables as described in [34]. Overdispersion was taken into account by observation-level random effects (OLRE) [32]. Models were simplified based on the Akaike information criterion (AIC) and the best-fitted models are reported. The significance of random effects in GLMM was determined by comparing reduced models (the random effect of interest dropped out) to full models using the anova function. All *p*-values presented are from two-tailed tests, with α = 0.05.

First, the effect of SP treatment on sperm swimming velocity (VCL) in relation to the control treatment (control vs. SP) was tested with LMM as follows: sub-sample (A and B), treatment (control or SP), time point, and time point–treatment interaction (time point × treatment) were used as fixed effects, and sperm donor, SP donor, and their interaction as random factors. To consider repeated measures of VCL at two time points, a random slope of time point was included in the final random effect terms (time point|sperm donor, time point|SP donor, time point|sperm donor:SP donor). An equivalent model was used in GLMM to test the effect of time and treatments on the proportion of motile cells and the proportion of hyperactivated sperm.

After these analyses, the control samples were filtered out from the dataset to focus only on the SP-treated sperm data [35]. The effect of each male’s own SP in relation to foreign SPs on the three above-mentioned sperm parameters was tested with similar LMM and GLMM models except that in these models the treatment included either own or foreign SP. When the interaction between treatment and time point was significant, we conducted similar analyses separately for each time point (30 and 90 min). These time-point-specific models included fixed effects of sub-sample (A and B) and SP treatment (own or SP) as well as random effects of the sperm donor, SP donor, and their interaction (1|sperm donor, 1|SP donor, 1|sperm donor:SP donor). Because sperm viability was measured only at one time point, the models for sperm viability were identical to these time-point-specific models.

The effect of three SP protein levels (APN, DPP4, and NEP) on the four sperm parameters were tested using otherwise equivalent LMMs and GLMMs as above, but protein level (covariate) and its interaction with time point were added as additional fixed factors (the effect of each protein was tested in separate models). If the protein level and its interaction with time point were statistically significant, we conducted separate analyses for both time points. Before visualization, VCL data were scaled to have a mean of 0 with a standard deviation of 1 to enable visual male-to-male comparison. Results were visualized using ggplot2 [36].

## 3. Results

### 3.1. Effect of Different SP Treatments on Human Sperm Performance

When comparing control (SW) samples with SP-treated samples, sperm swimming velocity (VCL) and the proportions of motile and hyperactivated (HA) sperm are affected by the treatment and the interaction between time point and treatment, indicating that the effect of treatment differs between time points (Appendix A). Time-point-specific analyses show that VCL (30 min: estimate = 42.69, t = 5.07, *p* = 0.001; 90 min: estimate = 59.16, t = 14.25, *p* < 0.001) and the proportions of motile and hyperactivated sperm (motility 30 min: estimate = 0.19, z = 4.09, *p* < 0.001; 90 min: estimate = 0.40, z = 4.62, *p* < 0.001; HA 30 min: estimate = 2.83, z = 3.00, *p* = 0.003; 90 min: estimate = 4.77, z = 5.91, *p* < 0.001) are significantly higher in the control samples than in SP-treated sperm at both time points (Figure 1, Appendix A). However, no difference in sperm viability was observed between control and SP-treated samples (estimate = 0.07, z = 0.72, *p* = 0.471).

When comparing own SP and foreign SPs, VCL was similar in both SPs (*p* = 0.973), and no interaction between time point and treatment is observed (*p* = 0.583) (Appendix A). The proportion of motile sperm and the proportion of hyperactivated sperm are also similar in both SPs (motility: *p* = 0.532, HA: *p* = 0.613), but these effects differed between time points (motility: SP treatment × time: *p* = 0.016; HA: SP treatment × time: *p* = 0.033). Time-point-specific analyses show no difference in the proportion of motile sperm at 30 min or the proportions of hyperactivated sperm at 30 min and 90 min (Table 1 and Table 2), but at 90 min, a higher proportion of sperm was motile in their own-SP-treated samples compared with foreign-SP-treated samples (Table 1). This difference is also clearly visually detectable from Figure 1. Sperm viability shows no difference between own- and foreign-SP-treated samples (Table 3).

When focusing only on the SP-treated sperm, we observed that all sperm parameters are significantly affected by time (Appendix A). Time-point-specific analyses reveal that sperm donor affected VCL, and the proportions of motile and hyperactivated sperm at both time points (Table 1, Table 2 and Table 4). In other words, males differed from each other based on their overall sperm performance (Figure 2, Appendix A). SP donor affected VCL and the proportion of hyperactivated sperm at both time points, whereas the proportions of motile sperm were affected by SP donor only at the 90 min time point. Sperm donor–SP donor interaction (combination) affected only VCL and the percentage of hyperactivated sperm and only at the 90 min time point. Sperm viability was affected by sperm donor, SP donor, and their interaction (Table 3, Appendix A). Sperm motility and viability were thus strongly dependent on both sperm donor identity and SP donor identity, and especially at the 30 min time point some males’ SPs tended to have a better overall ability to maintain VCL than others (e.g., males 4 and 6, indicated as purple and yellow box plots, Figure 2). It is interesting to note that sperm of male 3 has the lowest VCL at both time points (Appendix A), but SP of this male appeared to often be beneficial to foreign males’ sperm (green box plots in Figure 2). Moreover, sperm of male 3 sustained its VCL at a similar level between the 30 min and 90 min time points regardless of the SP the sperm were exposed to, whereas a clear reduction in VCL was observed in the sperm of males 1, 8, and 9 between time points (Appendix A). Contrary to male 3, male 9 had the highest VCL at 30 min but the effects of his SP on VCL were below or close to average (grey box plots in Figure 2, Appendix A). Such above-mentioned general effects of particular SPs were absent in the proportion of motile cells (Appendix A), i.e., no SP maintained motility across most of the males. Sperm hyperactivation shows a great degree of variation both within and between males (Appendix A). Furthermore, SPs of males 2 and 7 seem to have adverse effects on sperm viability in most cases (blue and brown bars in Appendix A). Finally, our results indicate that sperm viability and to a lesser degree also sperm VCL and hyperactivation were dependent on the identity of both the ‘receiving’ sperm and the SPs that were used to treat the sperm (i.e., sperm donor–SP donor interaction).

### 3.2. The Effect of SP Proteins on Sperm Swimming Velocity

The presence of DPP4, NEP, and APN was detected in each of the nine SP samples (Figure 3) in line with other studies [21,22,23,37,38]. Relative expression levels varied between 6.46–18.26 for DPP4, 0.70–3.50 for NEP, and 2.15–9.19 for APN (arbitrary unit values). The overall models (including both time points) show that VCL was positively affected by DPP4 and NEP (DPP4: estimate = 1.41, t = 2.65, *p* = 0.033; NEP: estimate = 7.43, t = 4.68, *p* = 0.002), but not by APN (estimate = 1.16, t = 0.85, *p* = 0.42). Furthermore, the effect of NEP differed between time points (NEP × time: estimate = −4.47, t = −3.47, *p* = 0.010), but for DPP4 and APN, the association was consistent for both time points (DPP4 × time: estimate = −0.64, t = −1.47, *p* = 0.184; APN × time: estimate = −0.62, t = −0.68, *p* = 0.517). Time-point-specific models reveal that a higher level of NEP was associated with higher VCL at both time points (NEP-30 min: estimate = 7.38, t = 4.67, *p* = 0.002; NEP-90 min: estimate = 2.95, t = 2.67, *p* = 0.032, Figure 4). In other words, DPP4 and NEP levels in SP were positively associated with sperm VCL at both time points.

### 3.3. The Effect of SP Proteins on Sperm Hyperactivation

The overall models (including both time points) reveal that the proportion of hyperactivated sperm was positively affected by the abundance of DPP4 and NEP in SP (DPP4: estimate = 0.16, z = 187.70, *p* < 0.001; NEP: estimate = 0.91, z = 4.55, *p* < 0.001), but not by APN (APN: estimate = 0.13, z = 0.80, *p* = 0.422). The effect of DPP4 on sperm hyperactivation differed between time points (DPP4 × time: estimate = −0.004, z = −4.17, *p* < 0.001), but was similar for NEP and APN (NEP × time: estimate = −0.11, z = −0.59, *p* = 0.558; APN × time: estimate = −0.03, z = −0.38, *p* = 0.743). Time-point-specific analyses reveal that DPP4 was associated with a higher proportion of hyperactivated sperm at both time points (30 min: estimate = 0.17, z = 2.54, *p* = 0.011; 90 min: estimate = 0.18, z = 2.69, *p* = 0.007). In other words, such as for sperm VCL (see above), DPP4 and NEP levels in SP were also positively associated with sperm hyperactivation at both time points.

### 3.4. The Effect of SP Proteins on the Proportion of Motile Sperm and Sperm Viability

The overall models (including both time points) show that the proportion of motile sperm cells was not affected by the abundance of any of the proteins measured (DPP4: estimate = 0.002, z = 0.94, *p* = 0.347; NEP: estimate = −0.01, z = −0.75, *p* = 0.451; APN: estimate = −0.01, z = −0.64, *p* = 0.525). However, the effect of NEP on the proportion of motile sperm differed between time points (NEP × time: estimate = −0.05, z = −2.36, *p* = 0.018), whereas for DPP4 and APN, the effect was consistent for both timepoints (DPP4 × time: estimate = −0.004, z = −0.43, *p* = 0.667; APN × time: estimate = −0.01, z = −0.99, *p* = 0.321). Time-point-specific models reveal that NEP abundance was associated with the proportion of motile sperm at 90 min, but not at 30 min (NEP 30 min: estimate = −0.01, z = −0.69, *p* = 0.489; NEP 90 min: −0.07, z = −2.77, *p* = 0.006). Sperm viability was not affected by the expression of any of the three proteins (DPP4: estimate = −0.01, z = −1.87, *p* = 0.062; NEP: estimate = 0.04, z = 1.01, *p* = 0.312; APN: estimate = 0.03, z = 1.78, *p* = 0.076).

### 3.5. Functional Enrichment Analysis of the Interactive Proteome of DPP4, NEP, and APN

To further clarify the biological function of the studied proteins, we searched their interactive proteome that fulfilled high confidence criteria using Search Tool for the Retrieval of Interacting Genes/Proteins (STRING) (Figure 5). Gene ontology (GO) analysis demonstrated that these interactive proteome members are enriched among immune response and apoptosis-related GO terms, such as positive regulation of fc receptor mediated stimulatory signaling pathway, negative regulation of anoikis (one form of programmed cell death), and regulation of b cell apoptotic process (Figure 5A). GO analysis also revealed several GO terms related to biological processes mediated by integrins and demonstrated that both DPP4 and NEP indirectly interact with the enriched proteins integrin subunit alpha 4 (ITGA4), integrin subunit alpha 5 (ITGA5), integrin subunit alpha V (ITGAV), and integrin subunit beta 1 (ITGB1) (Figure 5B). Furthermore, GO analysis illustrated that ITGB1 had a direct association with DPP4 based on “textmining” (light green) and “co-expression” (dark gray) evidence.

## 4. Discussion

Our results show that, when mixing sperm and SP from different males, sperm motility and viability strongly depended on the identity of the SP donor and in some cases also on the combination of SP donor and sperm donor. However, the effect of males’ own SP on sperm motility or viability was similar to that of foreign SP, although the proportion of motile sperm was higher in own SP after 90 min of SP treatment. To gain a greater insight into the factors that could drive the observed effects, we examined the abundance of three SP proteins (NEP, DPP4, and APN) and discovered that the abundance of two of them (NEP and DPP4) was positively associated with sperm swimming velocity and the proportion of hyperactivated sperm at both time points. Additionally, a higher abundance of NEP was associated with a higher proportion of motile sperm cells at 90 min. In summary, these findings indicate that variation both in the composition of the SP as a whole and its specific proteins can have an important role in determining sperm function. Therefore, a more detailed characterization of the exact SP components and their interactions responsible for the observed effects could potentially reveal novel biomarkers for male infertility.

The demonstrated SP-mediated effects on sperm traits are in line with previous findings in stallions and in rams [10,11]. Nevertheless, to the best of our knowledge, the present study represents the first attempt to investigate such effects in humans. Our results reveal that sperm motility in SP was consistently lower in SP-treated samples than in control samples, but no difference between treatment types was found in sperm viability. This indicates that SP treatment was not detrimental to the sperm, but it tends to inhibit sperm motility, as has been earlier demonstrated in humans [39]. Interestingly, the strength of the inhibitory effect differed across SP donors, but virtually no differences were found between own and foreign SPs (except for the proportion of motile cells at 90 min). It has been envisaged that SP could function as a spermicide against foreign sperm and that, e.g., seminal plasma cytokines could play an important role in such selective elimination of the foreign sperm [40]. On the other hand, den Boer et al. demonstrated that foreign seminal fluid decreased sperm viability only in polyandrous species, i.e., in animals where sperm from multiple males frequently co-exist in the female reproductive tract (sperm competition) [41]. However, such an inhibitory effect was absent in monandrous species, where females mate only with a single male. The human mating system is commonly regarded as largely (but not fully) monogamous (e.g., [42]), which could at least partly explain why we did not find strong evidence for the inhibition of foreign sperm by SP.

Our results also show that sperm performance is strongly dependent on the identity of the SP donor, indicating that some males’ SP may be consistently better in maintaining sperm function than others. On the other hand, we also found evidence that the sperm of some males might be relatively unresponsive to SP treatments (e.g., male 3, Figure 1). In any case, observed differences between SPs may result from sample-specific variation in the number of different SP components. Indeed, seminal plasma is a complex mixture of molecules produced by the accessory sexual glands: seminal vesicles, prostate, bulbourethral glands, vas deferens, and epididymides [9]. Therefore, our findings can potentially be explained at least by (a) variation in the relative proportions (quantity) of different accessory sex gland secretions, (b) alteration of the molecular composition of any of these fluids, (c) or both [11]. Since proteins constitute the largest fraction of the seminal plasma and are regarded as a key modulator of sperm function [43], it seems likely that the above-mentioned differences between SP donors could be largely explained by sample-specific variation in SP proteome.

Supporting this view, the abundance of two (out of three) SP proteins (NEP and DPP4) predicted sperm performance in our SP treatments, indicating that these SP peptidases are important regulators of sperm function. The observed positive association between NEP levels and sperm swimming velocity is in agreement with some earlier studies, which have demonstrated that NEP regulates the levels of enkephalins (by degrading them) that are known to inhibit sperm motility via activation of the opioid system in the sperm [44,45]. In other words, NEP degrades signaling molecules that are responsible for the activation of a signal cascade eventually resulting in sperm motility inhibition. The other protein, DPP4, is known to function during spermatogenesis, and also participates in boosting sperm mitochondrial metabolism [46,47], i.e., energy production crucial for sperm motility. Importantly, a lower abundance of DPP4 in SP has been linked to decreased sperm motility in males suffering from type 2 diabetes mellitus; thus, potentially explaining the decline of male fertility commonly observed in diabetic patients [47].

Finally, due to the complexity of the SP proteome, it is likely that in addition to NEP and DPP4, sperm performance is affected by a wide array of other SP proteins, such as those highlighted among the interactive proteome of studied proteins (see Figure 5), such as the integrin superfamily members (ITGA4, ITGA5, ITGAV, and ITGB1). Furthermore, heat shock protein family A member 2 (HSPA2), annexin A2 (ANXA2), sorbitol dehydrogenase (SORD), kallikrein-related peptidase 2 KLK2, and beta-tubulin, that have earlier been shown to be differentially expressed between normozoospermic and asthenozoospermic males, would be other potential candidates for future studies [48]. Furthermore, we advocate that the control of sperm performance by SP proteins potentially involves a complex orchestra of other biochemical factors, and identification of all these players would require mechanistic studies including the proteins themselves, their substrates, the receptors where the substrates act, and finally, the components of the downstream signaling cascade towards the observed effects. Therefore, we need much more experimental studies aiming to identify such factors and their relative importance as regulators of sperm function. Together, these studies could significantly facilitate the establishment of reference values for the most important SP factors responsible for sperm fertilization capability. This in turn could enable more accurate male infertility diagnostics and open novel possibilities to treat male infertility problems related to reduced semen quality, such as azoospermia (see e.g., [49]). However, since our data is based on a limited number of male subjects (*n* = 9), we encourage future studies to replicate our experiment in a larger number of males.

In conclusion, we demonstrate that the function of human sperm depends on the identity of SP, but that males’ own SP is not consistently better in maintaining sperm function than foreign SPs. These results indicate that the allospermicidal function of SP against foreign males’ sperm is absent in humans, but that male-specific differences in SP composition, however, can significantly modify sperm motility and viability. Furthermore, we show that the abundance of two SP proteins, NEP and DPP4, predicted sperm swimming velocity and hyperactivation. Although the underlying mechanisms and interactive players of these proteins are incompletely understood in the context of sperm functionality, our results shed light on the potential mechanistic processes driving the SP-mediated regulation of sperm performance. To facilitate the development of more accurate diagnostic approaches and better therapeutic interventions for male-factor infertility, we encourage further studies to identify other functionally important SP proteins and to evaluate their potential as biomarkers of seminal plasma quality and male fertility.

## Figures and Tables

**Figure 1 cells-11-02147-f001:**
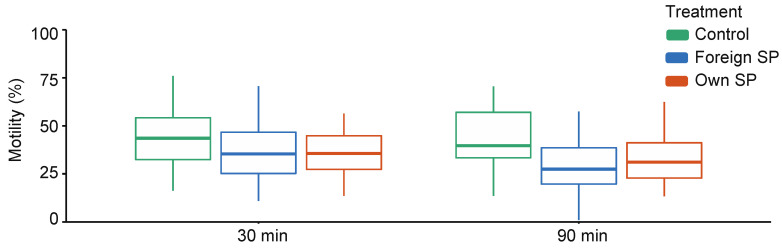
Sperm swimming velocity in different sperm treatments (control, foreign seminal plasma, and own seminal plasma) at both time points (30 min and 90 min). Bolded horizontal lines in each box plot denote median values and boxes extend from the 25th to the 75th percentile of the distribution of values. Vertical lines show the minimum and maximum values of the data. SP = seminal plasma.

**Figure 2 cells-11-02147-f002:**
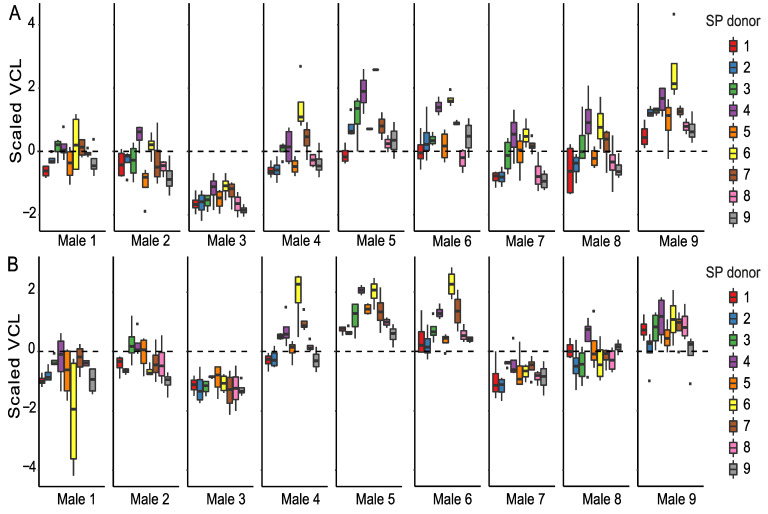
Sperm swimming velocity (VCL) in SP-treated sperm samples. Sperm VCL is scaled to a mean of zero (dashed line, see methods for details) to enable male-to-male comparison at 30 min (**A**) and 90 min (**B**) time points. Each male (1–9) is represented in a separate panel and the SPs used for sperm treatments are indicated by different colors.

**Figure 3 cells-11-02147-f003:**
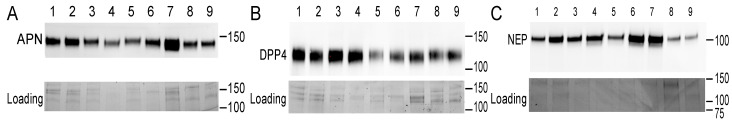
Presence of proteins of interest (above) and their loading controls (below) in seminal plasma samples. Western blot was used to determine the level of APN (**A**), DPP4 (**B**), and NEP (**C**) (males indicated by numbers 1–9). The position of the molecular weight standard (kDa) is shown on the right side of each panel. The blots shown here represent two other replicates based on a visual inspection of all the blots (*n* = 3 replicates, in total).

**Figure 4 cells-11-02147-f004:**
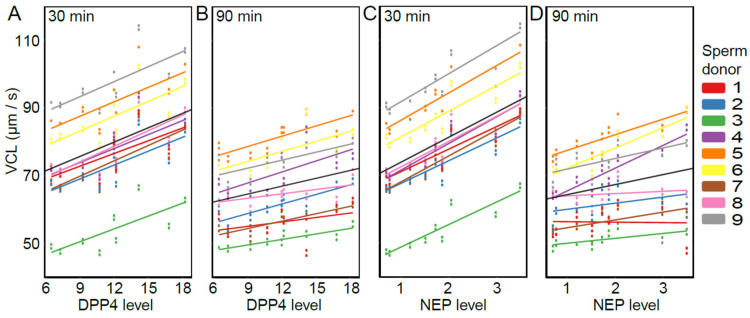
The association between sperm swimming velocity (VCL) and the abundance of DPP4 and NEP in SP samples: DPP4 30 min (**A**), DPP4 90 min (**B**), NEP 30 min (**C**), and NEP 90 min (**D**). Data points represent fitted values obtained from the LMM. Sperm donors are indicated by different colors and the black line represents the average slope across all males. Levels of DPP4 and NEP (x-axis) are shown as arbitrary unit values.

**Figure 5 cells-11-02147-f005:**
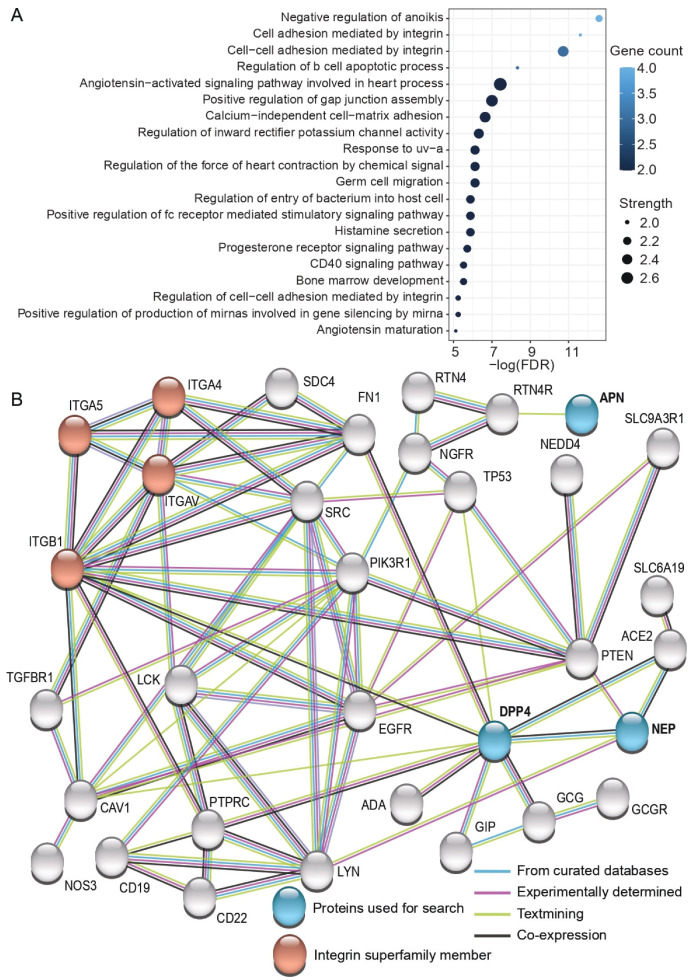
Functional enrichment (**A**) and network map (**B**) of the interactive proteome of DPP4, NEP, and APN in humans. Interactive proteins are based on the Search Tool for the Retrieval of Interacting Genes/Proteins (STRING) with high confidence interactions (see methods for details). Proteins inputted to the tool are indicated in blue and the proteins belonging to the integrin superfamily are marked in red. The source of interactions is indicated using colored lines: light blue stands for “curated databases”, purple refers to “experiments”, light green indicates “textmining”, and dark gray implies “co-expression” of proteins. FDR = false discovery rate.

**Table 1 cells-11-02147-t001:** Generalized linear mixed model (GLMM) statistics for the effect of sperm donor, SP donor, and their interaction (random effects) on the proportion of motile sperm cells at two time points (30 and 90 min) after the initiation of the SP treatment. Models also included fixed effects of sub-sample and treatment (own or foreign SP). OLRE = observation-level random effect.

Effects	% of Motile Sperm Cells 30 min				% of Motile Sperm Cells 90 min			
Random		d.f.	*χ^2^*	*p*		d.f.	*χ^2^*	*p*
Sperm donor		1	156.33	**<0.001**		1	102.36	**<0.001**
SP donor		1	3.39	0.066		1	8.35	**0.004**
Sperm donor x SP donor		1	0	1		1	2.33	0.127
OLRE		1	2.27	0.131		1	20.24	**<0.001**
**Fixed**	**Estimate**	**SD**	**z**	** *p* **	**Estimate**	**SD**	**z**	** *p* **
Intercept	−1.03	0.11	−9.38	**<0.001**	−1.26	0.11	−11.24	**<0.001**
Sub-sample	−0.04	0.02	−1.76	0.079	−0.08	0.03	−3.00	**0.003**
Treatment	−0.03	0.04	−0.71	0.478	0.11	0.05	2.26	**0.024**

**Table 2 cells-11-02147-t002:** Generalized linear mixed model (GLMM) statistics for the effect of sperm donor, SP donor, and their interaction (random effects) on the proportion of hyperactivated sperm cells at two time points (30 and 90 min) after the initiation of the SP treatment. Models also included fixed effects of sub-sample and treatment (own or foreign SP). OLRE = observation-level random effect.

Effects	% of Hyperactivated Sperm Cells 30 min				% of Hyperactivated Sperm Cells 90 min			
Random		d.f.	*χ^2^*	*p*		d.f.	*χ^2^*	*p*
Sperm donor		1	63.19	**<0.001**		1	18.77	**<0.001**
SP donor		1	59.15	**<0.001**		1	7.63	**0.006**
Sperm donor x SP donor		1	0	1		1	4.74	**0.030**
OLRE		1	24.17	**<0.001**		-	-	-
**Fixed**	**Estimate**	**SD**	**z**	** *p* **	**Estimate**	**SD**	**z**	** *p* **
Intercept	−5.41	0.53	−10.10	**<0.001**	−7.23	0.59	−12.20	**<0.001**
Sub-sample	−0.12	0.14	−0.80	0.418	−0.35	0.23	−1.52	0.126
Treatment	−0.09	0.23	−0.41	0.684	0.82	0.44	1.84	0.065

**Table 3 cells-11-02147-t003:** Generalized mixed model (GLMM) statistics for the effect of sperm donor, SP donor, and their interaction (random effects) on sperm viability. Model also included fixed effects of sub-sample and treatment (own or foreign SP). OLRE = observation-level random effect.

Effects	% of Viable Sperm Cells			
Random	d.f.		*χ^2^*	*p*
Sperm donor	1		98.98	**<0.001**
SP donor	1		71.81	**<0.001**
Sperm donor × SP donor	1		15.57	**<0.001**
OLRE	1		789.33	**<0.001**
**Fixed**	**Estimate**	**SD**	**z**	** *p* **
Intercept	−1.50	0.06	−26.15	**<0.001**
Sub-sample	0.01	0.01	0.91	0.363
Treatment	−0.04	0.03	−1.84	0.066

**Table 4 cells-11-02147-t004:** Linear mixed model (LMM) statistics for the effect of sperm donor, SP donor, and their interaction (random effects) on sperm swimming velocity (VCL) at two time points (30 and 90 min) after the initiation of the SP treatment. Models also included fixed effects of sub-sample and treatment (own or foreign SP).

Effects	Sperm VCL 30 min					Sperm VCL 90 min				
Random			d.f.	*χ^2^*	*p*			d.f.	*χ^2^*	*p*
Sperm donor			1	117.77	**<0.001**			1	81.59	**<0.001**
SP donor			1	69.53	**<0.001**			1	11.74	**0.001**
Sperm donor x SP donor			1	1.52	0.217			1	22.31	**<0.001**
**Fixed**	**Estimate**	**SD**	**d.f.**	**t**	** *p* **	**Estimate**	**SD**	**d.f.**	**t**	** *p* **
Intercept	79.85	5.07	13	15.72	**<0.001**	66.74	3.77	10	17.72	**<0.001**
Sub-sample	−1.18	0.94	235	−1.26	0.210	−1.56	0.79	241	−1.96	0.051
Treatment	0.02	1.68	62	0.01	0.992	1.14	1.95	63	0.59	0.559

## Data Availability

The data presented in this study are openly available in the GitHub Repository at: https://github.com/tanjahturunen/Seminal_plasma_-_mediated_effects_on_sperm_swimming_performance_in_humans (accessed on 15 June 2022).

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
