# Peer review of "Seminal-Plasma-Mediated Effects on Sperm Performance in Humans"

_cells, 2022, doi:10.3390/cells11142147_

Round 1

Reviewer 1 Report

The study is well designed  but I recommend some minor revision in order to be published..... what about ethical consent for participation for patients????......many references are old not relevant.... all references before 2005 must be removed and replaced by new ones......figure 1 and 2 with very bad resolution ...please clarify......aim of the work should be rephrased..

.first paragraph in discussion should be rewritten..... conclusion section should be summarized

Reviewer 2 Report

In this interesting study the effect of among-male variation in human seminal plasma (SP) on sperm motility and viability was examined. The addition of SP reduced sperm motility but not sperm viability. However, the degree of reduction in sperm motility depended on the identity of the donor and recipient. Next, the abundance of three SP proteins was correlated with sperm traits, and two (DPP4 and NEP) were found to have a positive relationship with sperm velocity, providing a possible mechanism for observed changes in this trait but not other traits. Overall, the results are certainly intriguing. More information would help the reader assess and interpret the findings (see below).

Major comments:

The results are quite difficult to follow. It would be good to include a figure of the treatment effects to help the reader assess and interpret effects of control versus own and foreign SP. In particular, I am curious whether the effect of own vs foreign SP on sperm motility was trending in the same direction at 30mins but only reached significance at 90mins.

It would be good to unpack the interaction (or lack thereof) between sperm donor and recipient in more detail. In particular, I am interested in the correlation between the sperm quality of the semen sample and the effect that the seminal plasma had on competitor sperm. Although there is variability, most samples show a domed shape pattern at 30min in Fig 1, suggesting that the SP samples have fairly consistent effects across males. However, what is striking to me is that the sperm velocity of donor 3 is the lowest (Fig 3), yet the SP of donor 3 seems to generally support above average sperm velocities in competitor samples (Fig 1). Conversely, donor 9 had the fastest swimming sperm, yet the sperm velocities in SP from donor 9 were generally below average. Of course, the sperm velocities of sample 1 (used to obtain SP) may differ significantly from sample 2. Did you measure the sperm motility and velocity of the first samples? It would be very interesting to explore this relationship.

Are there any correlations between the effect of SP on sperm motility and sperm viability? The only data we can see is scaled VCL, so we cannot assess whether certain SP samples had overall positive effects on semen quality (ie/ increased % motility and viability), or opposing effects etc.

Minor comments/questions:

L228-233: Is it normal for human sperm samples to increase in % motility and velocity from 30 min to 90 min?

Were all semen samples above WHO guidelines? What was the range in % motility and % viability?

Reviewer 3 Report

In this study, Turunen et al. address the potential beneficial roles of of seminal plasma addition to the resulting selected sperm vitality characteristics. The topic of the roles of seminal plasma in the sperm behavior are a hot topic in modern andrology, and the study presents with novel and interesting data. I also appreciate the straight-forwardenss and clarity of the manuscript.

I have several minor comments:

- The sample size is quite limited. As such, I would recommend to discuss this issue as a limitation of the study.

- what were the exclusion/inclusion criteria for the subjects included in the study?

- what other proteins would be worth further assessment besides DPP4, NEP and APN? Are there any suggestions for a potential synegry or antagonism of these within the heterogenous seminal plasma?

- please explain the abbreviations when first used /such as in the Abstract section)

- Please, carefully correct the references within the main body (e.g. page 2, line 58)

- why DNA fragmentation was not assessed in the samples? It is known by now that the smeinal plasma might possess a nuclease activity that might compromise the sperm DNA integrity.
